# Genome-Wide Identification and Characterization of the Oat (*Avena sativa* L.) WRKY Transcription Factor Family

**DOI:** 10.3390/genes13101918

**Published:** 2022-10-21

**Authors:** Kaiqiang Liu, Zeliang Ju, Zhifeng Jia, Guoling Liang, Xiang Ma, Wenhui Liu

**Affiliations:** Key Laboratory of Superior Forage Germplasm in the Qinghai-Tibetan Plateau, Qinghai Academy of Animal Husbandry and Veterinary Sciences, Qinghai University, Xining 810016, China

**Keywords:** *Avena sativa* L., WRKY family, genome-wide, expression patterns

## Abstract

The WRKY family is widely involved in the regulation of plant growth and stress response and is one of the largest gene families related to plant environmental adaptation. However, no systematic studies on the WRKY family in oat (*Avena sativa* L.) have been conducted to date. The recently published complete genome sequence of oat enables the systematic analysis of the AsWRKYs. Based on a genome-wide study of oat, we identified 162 AsWRKYs that were unevenly distributed across 21 chromosomes; a phylogenetic tree of WRKY domains divided these genes into three groups (I, II, and III). We also analyzed the gene duplication events and identified a total of 111 gene pairs that showed strong purifying selection during the evolutionary process. Surprisingly, almost all genes evolved after the completion of subgenomic differentiation of hexaploid oat. Further studies on the functional analysis indicated that AsWRKYs were widely involved in various biological processes. Notably, expression patterns of 16 AsWRKY genes revealed that the response of AsWRKYs were affected by stress level and time. In conclusion, this study provides a reference for further analysis of the role of WRKY transcription factors in species evolution and functional differentiation.

## 1. Introduction

Plants and animals are subjected to different evolutionary stresses in nature because plants are incapable of escaping predation and environmental changes [1], and in turn have reprogrammed their physiological, biochemical, and morphological features to tolerate or adapt to the current environment [2]. This process requires numerous genes and extensive signal coordination [3]. Interestingly, plant transcription factors (TFs) are important in regulating spatial gene expression and signal transduction. They also act as a self-regulatory network that assists plants in adapting to the ever-changing environment. Therefore, investigating TFs can provide some insights into the genetic and evolutionary history of plants. The WRKY gene family is one of the largest transcription factor families in higher plants, and multiple members of the WRKY transcription factor have been found in all living land plants [4]. These play an essential role in plant growth, development, and defense against biotic and abiotic stresses [5].

Since the first WRKY sequence discovery in sweet potato (*SPF1*) [6], wild oat (*ABF1,2*) [7], and Arabidopsis (*ZAP1*) [8], accumulated research evidence has demonstrated that the functions of the WRKY family influence almost all physiological and metabolic processes in plants such as hormone pathways, signal transduction, growth, and development. However, our analysis of the function of related genes in different WRKY deletion or overexpression mutants revealed that the role of WRKY TFs in plant defense response was relatively complex. For example, the loss of WRKY in *stp* mutants led to the derepression of *NST2* and C3H zinc-finger transcription factors, activating downstream gene expression and thereby increasing stem biomass [9]; and the mutant for *atwrky62* negatively regulated the plant growth [10]. In addition, WRKY TFs also have positive and negative regulatory roles in biotic and abiotic stresses. In Arabidopsis, *AtWRKY8 and AtWRKY28* positively enhanced resistance to *Botrytis cinerea*, and *AtWRKY8* and *AtWRKY48* did not positively enhance resistance to *Pseudomonas syringae* [11,12,13]. In rice, *OsWRKY08* [14], *OsWRKY11* [15], *OsWRKY23* [16], *OsWRKY45* [17], *OsWRKY72* [18], and *OsWRKY89* [19] positively or negatively regulated plant responses to cold, drought, hormones, or salt. Interestingly, WRKY genes do not always directly act on target genes, and the same gene may respond to multiple stresses [20]. In wheat, *TaWRKY2* and *TaWRKY19* combined with *RD29B, RD29A, RD29B*, or *COR6.6*, increased the expression of downstream genes under drought stress [21]; *TaWRKY51* promoted lateral roots by negatively regulating ethylene biosynthesis [22]; overexpression of *TaWRKY146/2/19* both enhanced the resistance of salt or drought [5,23]. There was also growing evidence for WRKY protein interactions. Eulgem et al. [24] showed that the homology of *AtWRKY18*, *AtWRKY40*, and *AtWRKY60* were relatively high, which not only formed three different homologous complexes through their own interaction, but also formed heterologous complexes through their interactions with each other. Hwang et al. [25] found that the synergistic interaction between *OsWRKY51* and *OsWRKY71* inhibited the signal of gibberellic acid (GA) in aleurone cells of rice seeds.

Some WRKY proteins may contain multiple domains, whereas others contain only highly conserved WRKY DNA-binding domains, which are a common feature of WRKY TFs [26,27,28]. The most prominent features of the WRKY family include an N-terminus that harbors a WRKYGQK heptapeptide sequence domain and the C-terminus is a zinc finger structure composed of C-X_4–5_-C-X_22–23_-HXH (C_2_H_2_) or C-X_7_-C-X_23_-HXC (C_2_HC) [5]. The WRKY protein generally has one or two WRKY domains and based on its number can be classified into three groups (I–III). Group I proteins contain two WRKY domains and a C_2_H_2_ motif; group II and III proteins only have a single WRKY domain and have a distinct zinc finger motif. The former has the C_2_H_2_ motif and is further divided into five subgroups (IIa-e) [5,26], whereas the latter has the C_2_HC motif. Interestingly, group I proteins are not restricted to the plant lineage and were also found in the unicellular eukaryote *Giardia lamblia* and slime mold *Dictyostelium discoideum* [29,30,31], but group II and III proteins are stably present in the green plant lineage. In addition, the promoter regions of target genes regulated by WRKY contained a W-Box or WT-Box, and are highly conserved [32]. Among these, either TGAC or GACTTTT motif is the core sequence of the W-box and is highly conserved [32]. Once a nucleotide in core motifs are changed, the ability of the WRKY protein to bind to it will be affected or even lost [33].

The WRKY family already evolved during the moss and fern era [34]. Although WRKY proteins were considered as plant-specific, the monophyletic WRKY domain can be traced back to the original WRKY protein with only one WRKY domain, which was presumed to exist in basic non-plant eukaryotes [30]–such as group I WRKY–and was not limited to plant lineages. The process of modern angiosperms evolution–especially in grasses and asterids–and the continuous stimulation of environmental change caused variations and increased the number of families of different types of land plants [30,35]. Gene duplication events are the main method responsible for the expansion of WRKY genes and are also the driving force for generating new genes and diversity. For instance, the N and C terminal WRKY domains have a high degree of structural similarity, which was caused by domain replication events during the early evolution of the WRKY protein [27]. In addition, another study has shown that 80% of WRKY loci in rice are located in duplicated regions [36]. Notably, WRKY proteins are conserved throughout the evolutionary history of the plant lineage, but there are still significant differences among species—71 WRKY genes were found in *Arabidopsis thaliana* [29], 171 in *Triticum aestivum* [37], 31 in *Oryza sativa* [38], 182 in *Glycine max* [39], and 119 in *Zea mays* [40]. Therefore, further elucidation of the evolution of the WRKY family requires additional investigations.

Oat (*A. sativa*) is an allohexaploid (*2n = 6*× *= 42*) plant with a large and complex genome and is one of the least explored in genome and transcriptome among cereal grain crops [41]. Oats can be grown alone or intergrown with annual legumes to provide dry and green fodder. In the cold temperate regions of the Northern Hemisphere, it is an important cultivated crop in Canada, Russia, and the Nordic countries [42]. Meanwhile, owing to its grain and high yield and quality, oat has been playing an outstanding role in livestock production in China. Moreover, oat has higher resistance to adverse conditions than rice, wheat, and other feed crops and can be used as a pioneer crop for soil improvement [43]. Given the economic and ecological value of oat, it is thus essential to identify important functional genes, and fortunately, the genome sequence of oat has recently been published (https://wheat.pw.usda.gov/GG3/graingenes-downloads/pepsico-oat-ot3098-v2-files-2021) (accessed on 3 March 2022). In this study, we identify WRKY TFs from oat and analyze features of AsWRKYs and expression profiles. The results will lay the foundation for further studies of WRKY TFs in other crops, including their evolutionary relationships and phylogeny.

## 2. Materials and Methods

### 2.1. Genome-Wide Identification of the WRKY Gene in Oat

In this study, the oat proteins sequence and HMM file of the WRKY domain (PF03106) were downloaded from the grain genes database (https://wheat.pw.usda.gov/) (accessed on 3 March 2022) [44] and Pfam database (https://pfam.xfam.org/) (accessed on 3 March 2022) [45], respectively. Based on the HMMER file to search, the WRKY genes in the oat proteins and the default parameter for cutoff E-value was 0.01. All candidate genes were further verified by the Pfam, CDD (https://www.ncbi.nlm.nih.gov/cdd/) (accessed on 3 March 2022) [46], and SMART (http://smart.embl.de/smart/batch) (accessed on 3 March 2022) [47] database to confirm all genes contained a conserved heptapeptide WRKYGQK and zinc-finger structure. Then, we further analyzed the Mw and pI of all identified genes by Expasy (https://web.expasy.org/compute_pi/) (accessed on 3 March 2022) [48]. Furthermore, the subcellular localization of these genes was also predicted with CELLO (http://cello.life.nctu.edu.tw/) (accessed on 3 March 2022) [49] and WoLF PSORT (https://www.genscript.com/wolf-psort.html) (accessed on 3 March 2022) [50]. The SSR site was predicted by SSRIT [51].

### 2.2. WRKY Classification, Phylogenetic Construction, Conserved Motifs, and Promoter Analysis of AsWRKY

The WRKY domain of AsWRKY proteins were subjected to multiple sequence alignments using MEGA 7. The amino acid sequences subjected to BLAST were manually adjusted for group classification using GeneDoc tools (ScienceOpen, Inc., Burlington, MA, USA). Then, phylogenetic reconstruction based on the WRKY domain of Arabidopsis and oat was performed using the neighbor-joining method, and the program parameters were as follows: p-distance, 80% cutoff of partial deletion, and 1000 bootstrap repeats. The conserved motifs of all WRKY proteins were analyzed using MEME (https://meme-suite.org/meme/tools/meme) (accessed on 3 March 2022) [52], and the default parameters were as follows: the number of motifs was 10, with the optimum motif width ranging from 6 to 50. The exon-intron structure of the AsWRKY gene was extracted from the genome annotation file using TBtools (v1.098769) software (Chengjie Chen, Guangzhou, China) [53]. To identify the *cis*-elements of the promoter, the upstream 1500 bp sequences of all AsWRKY genes were screened using PlantCare (https://bioinformatics.psb.ugent.be/webtools/plantcare/html/) (accessed on 3 March 2022) [54].

### 2.3. Chromosome Location and Collinearity Analysis of AsWRKY Genes

Based on the genome annotation file of oat, TBtools [53] was used to complete the chromosomal distribution and mapping of all genes. Gene duplication events were analyzed using the Multicollinearity Scanning Toolkit (MCScanX) with default parameters [55]. To demonstrate the synteny of orthologous WRKY genes obtained from oat with Arabidopsis, maize, rice, and wheat, a synteny analysis plot was constructed using TBtools (Dual Synteny Plotter). KaKs_Calculator 2.0 (https://github.com/kullrich/kakscalculator2, accessed on 3 March 2022)was used to calculate non-synonymous (Ka) and synonymous (Ks) substitutions for each duplicated gene pair [56]. Based on 2.6 × 10^−9^ substitution/synonymous site in oat [57], the approximate date of the repeat event (million years ago, Mya) was estimated using the formula T = Ks/2λ × 10^−6^.

### 2.4. Gene Ontology and miRNA-AsWRKY Prediction 

The Gene Ontology of oat WRKY proteins were obtained from eggNOG databases (http://eggnog-mapper.embl.de/) (accessed on 10 March 2022) [58]. The full-length protein sequences were upload to the eggNOG databases to obtain the gene annotation background file, and based on the TBtools software to screen the WRKY annotation information (molecular function, cellular component, and biological process) [53]. 

Based on the previously obtained miRNA data, 623 miRNAs were identified by sRNAminer [59]. The AsWRKY genes targeted by miRNAs were predicted using sRNAminer analysis software.

### 2.5. Transcriptome Expression Analysis of AsWRKY under Various Developmental Stages or Conditions

In this study, data SRP294982, SRP235083, SRP094911, SRP094513, and SRP237902 from NCBI (https://www.ncbi.nlm.nih.gov/sra/?term=oat) (accessed on 15 March 2022) were analyzed with Kallisto [60], and the AsWRKY gene expression profiles were calculated under lemma development, seed vigor, salt stress, phosphorus deficiency, and drought conditions [61,62,63]. In addition, we also analyzed transcriptome data under different drought conditions from our previous study. Transcripts per million (TPM) was used to calculate the gene expression and heat maps were produced using TBtools software [53].

### 2.6. Plant Material, Growing Conditions, Drought and Salt Treatments

Oat seeds were sown on soil in plastic pots and grown in a greenhouse. After 10 days, the seedlings were acclimated to growth chamber conditions for 2 days and further treated in 20% PEG6000 and 150 mM NaCl solutions for 0, 2, 4, 8, 12, and 24 h, respectively. Then, stressed seedlings were collected for RNA extraction and stored at −80 °C. 

### 2.7. RNA Isolation and Expression Analysis 

The total RNA of all samples was isolated using the plant RNA Kit (TAKARA, Xining, China), and then cDNA was synthesized by a PrimeScript™ RT reagent Kit (TAKARA, Xining, China). The 16 AsWRKYs were selected to validate the expression level under drought and salt stress and primers, as listed in Appendix A. qRT-PCR was performed on Light Cycle96 with TB Green^®^ Premix Ex Taq™ II (TAKARA, Xining, China), and the PCR program was conducted as follows: 10 s at 95 °C, 40 cycles of 95 °C for 5 s and 60 °C for 30 s. Each reaction had three biological replicates, and data were analyzed by the 2^−ΔΔCT^ method.

## 3. Results

### 3.1. Identification of the WRKY Family in Oat

Using the HMMER program, we searched for WRKY domains (PF03106) in the oat genome, and these were further checked for the conserved domain in the Pfam and CDD databases. Finally, a total of 162 candidate genes were identified; these encode 162 WRKY proteins. According to their order of chromosomal distributions, these genes were designated as *AsWRKY1* to *AsWRKY162* (Appendix A). Among these proteins, 32 annotations indicated that they each had two complete WRKY domains, 125 genes contained only one, and five genes lost their WRKY domain.

Gene signatures such as coding sequence (CDS) length, protein sequence length, protein molecular weight (MW), isoelectric point (pI), and subcellular localization were analyzed (Appendix A). There were obvious differences among the 162 AsWRKY proteins. For example, *AsWRKY132* has the greatest protein length (4494 aa) and MW (168.78 kDa), while the smallest one was *AsWRKY95* (399 aa and 14.46 kDa). The pI of proteins ranged from 4.61 to 10.32, which were identified in *AsWRKY158* and *AsWRKY156*, respectively. Further analysis using protein subcellular localization predictions revealed that all proteins, except for *AsWRKY2/7/19*, were located in the nuclear region. In addition, we also predicted the SSR site of AsWRKYs. In total, 92 putative new SSRs were detected from 62 sequences (Appendix A), which were mainly contained dinucleotide and trinucleotide. Amount them, more than half were in CDS regions, followed by exon and intron regions, and the main motif types were TC, CA, CAG, CGC, etc.

### 3.2. Multiple Sequence Alignment, Phylogenetic Analysis, and Classification of AsWRKY Proteins

To investigate the evolutionary relationship among members of the oat WRKY family, we selected Arabidopsis (*AtWRKY58, 40, 61, 28, 7, 14,* and *54*) and all oat WRKY domains for multiple sequence alignment. Alignment of the amino acid sequences found two significant regions, the first of which is conserved heptapeptide (WRKYGQK) in the WRKY domain that is the binding site of the W box (Figure 1). Various other WRKY domains were also found, including WHTYGQN, WRKHDEK, WRKYGEK, WRKYGHK, WRKYGKK, WRKYGKR, WRKYGQK, and WRYYGQK, and these were located in groups I, IIc, and III. Moreover, some WRKY domains were missing in groups IIc, IIe, and III. The second conserved domain was zinc finger structure that contains two types: C-X_3–5_-C-X_22–24_-H-X-H (C_2_H_2_) and C-X_4–7_-C-X_23–29_-H-X-C (C_2_HC). Among these, 112 AsWRKYs contained C_2_H_2_ zinc finger motifs, 46 AsWRKYs included C_2_HC, and the others (*AsWRKY28/95/100/103*) lost their zinc finger motifs.

A phylogenetic tree containing about 60 amino acids of the WRKY domains was constructed. According to previous studies on WRKY genes, their phylogenetic relationships, and WRKY domain characteristics of *A. thaliana*, the WRKY genes of oat and Arabidopsis could be divided into three groups–namely, I, II, and III (Figure 2). Group II was further divided into five subgroups (IIa, IIb, IIc, IId, and IIe). It is worth noting that *AsWRKY73* is clustered with group IN; they seem to have a more similar evolutionary relationship. Further analysis showed that *AsWRKY131* is highly homologous in terms of its zinc finger structure with group III; it is speculated whether *AsWRKY131* underwent transfer, mismatch, or recombination during chromosomal replication. Surprisingly, *AsWRKY86* and *AsWRKY92* belong to group IIc, although these were closer to group III. Is this related to the mutation of these two conserved domains? In contrast to group I, group II (83 AsWRKYs) contains a single WRKY domain. Thirteen members were found in subgroup IIa, 11 in IIb, 27 in IIc, 14 in IId, and 18 in IIe.

### 3.3. Gene Structure and Conserved Motifs Analysis of AsWRKY Genes

To further understand the relationship between gene structure and evolution among the AsWRKY genes, exon-intron distribution and conserved motifs were analyzed according to their full-length phylogenetic relationship. The number of introns in oats ranged from 0 to 9, with an average of 2.32 per gene (Figure 3B), of which *AsWRKY73/131/132* contain 9 introns and have the highest number of detected introns. It is worth noting that groups I and IIb have higher transcript abundance–the average number of introns is greater than 3.5 and was higher than that of other groups (Appendix A). Moreover, a total of 70 (43.2%) AsWRKY genes have two introns and 26 (16.0%) with three introns, which account for more than half of all genes. In terms of intron-exon conservation, it has the characteristics of group distribution. For example, the number of similar intron-exon structure in groups IIa, IIb, IId, IIe, and III was more than that of groups I and IIc. Interestingly, the intron phase was also highly conserved in all groups; we found a total of 293 phase 0, 203 phase 1, and 8 phase 2. It was worth noting that there were no phases 1 and phase 2 in IIa and IIb, and no phase 2 in IIe and III.

The conserved motifs of WRKY proteins were analyzed using the MEME program, and 10 motifs were identified in 162 AsWRKY proteins, with motif lengths ranging from 15 to 50 amino acids (Appendix A). As shown in Figure 3, within the same group, AsWRKY members usually have similar motifs composition that ranged from 1 to 7. Except for *AsWRKY36/67/73/80/95/132*, almost all proteins contain motifs 1 and 2 that define WRKY domains and zinc finger structures, respectively. However, some motifs belong to a special group, such as motif 3, 4, 7, and 8, which are only found in groups I and II or III. Notably, there are similarities among subgroups such as groups II a and II b, II d, and II e. These results indicate that proteins with the same or similar structures may be functionally or evolutionarily similar and demonstrate the reliability of the classification.

### 3.4. Chromosomal Location, Duplication, and Synteny Analysis of AsWRKY Genes

To further analyze the chromosomal distribution of AsWRKY genes, we mapped all genes based on the available gene annotation information in oat (Figure 4, Appendix A). The results showed that 162 genes could be mapped to 21 oat chromosomes, and the number of genes in the A, C, and D subgenomes (59, 46, and 57) did not significantly differ overall–but this was not the case for individual chromosomes. Among these, 5A (15, 9.26%), 1A (14, 8.64%), 3D (13, 8.07%), and 5D (13, 8.07%) showed relatively greater numbers distribution. In contrast, 6D has the lowest number distribution of genes (2, 1.23%), and interestingly 6D was also the shortest chromosome among them. Moreover, gene density was not positively correlated to the number of chromosomes, with AsWRKY gene density ranging from 0.004/Mb (4C) to 0.031/Mb (5A) (Appendix A).

Gene duplication provides raw materials and is the source of power for biological evolution [64]. Although there was no clear positive correlation between genome size and gene family size, we identified a large number of fragment duplications and tandem duplication events in allohexaploid oats. In total, we identified 108 pairs of segmental duplications (126 AsWRKY genes) and three tandem duplications (*AsWRKY100/101*, *AsWRKY116/117*, *AsWRKY122/123*, *and AsWRKY12/13*) (Figure 5). Notably, some AsWRKY genes have more than one repeat. In addition, we further calculated the non-synonymous replacement (Ka) and synonymous replacement (Ks) ratio of all AsWRKY gene pairs to determine which selection pressure affected gene evolution trends (>1, positive selection; =1 neutral selection; <1 purifying selection). The results indicated that all gene pairs had Ka/Ks values between 0 and 1. Specifically, the Ka/Ks values of 111 replication pairs corresponding to 129 WRKY genes were <1, indicating that these genes mainly underwent purifying selection, and no gene pair (Ka/Ks > 1) may have evolved under strong positive selection after replication. Then, to understand whether the evolutionary rate of the WRKY subgroup was identical, as shown in Appendix A, the average values of Ka/Ks were IIc > III > IIa > IIb > I > IIe > IId, which suggested the evolution rate was different among WRKY subgroups, and IIc was the fastest. Moreover, we also analyzed the approximate date of occurrence of duplication events. Duplication events in the AswRKY gene occurred between 0.18 Mya (Ks = 0.0047) and 41.16 Mya (Ks = 1.070) with an average of 6.26 Mya (Ks = 0.163) (Appendix A).

To further deduce the evolutionary mechanism of the oat AsWRKY gene family, we constructed a collinearity relationship with four representative model plants, such as Arabidopsis, maize, rice, and wheat (Figure 6, Appendix A). The number of syntenic genes for oat with other species was as follows: 231 orthologous pairs between 97 AsWRKYs and 116 TaWRKYs, 79 paired collinearity relationships between 76 AsWRKYs and 41 OsWRKY genes, 44 pairs between 41 AsWRKY and 28 ZmWRKYs, and only 5 pairs between Arabidopsis and oat. In total, we found 35 AsWRKY common genes in these collinearity relationships (except paired oat with Arabidopsis). Additionally, some AcWRKYs have been found to be associated with the presence of at least two or more syngenetic pairs (particularly between wheat and oat). For example, two or more homologous pairs had 78 genes (80.41%) in wheat, 2 (2.63%) in rice, and only 1 (2.4%) in maize, which indicated that these genes might have played an important role in evolution and were more frequently involved in gene duplication events.

### 3.5. Structural Analysis of Gene Promoters

To identify potential cis-acting elements for gene transcription initiation regulation, we analyzed the upstream 1500-bp promoter sequences of all WRKYs using the online PlantCARE site. Overall, 30 *cis*-acting elements were identified to be involved in abiotic and biotic stress, growth and development, and hormone signal transduction (Figure 7). AREB, TGACG, CGTCA, and G-box motifs are widely distributed in most AsWRKY genes and are mainly involved in responses to hormones and light. In addition, other common regulatory elements also accounted for a large proportion; for example, 98.7% of genes contained at least one element involved in hormone signal transduction and light responses. The relatively more enriched elements are AREB for ABA response, TGACG and CGTCA for jasmonic acid response, and TGA-element G-box, Sp1, and GT1-motif for light response. Approximately 74.7% of the genes had at least one element involved in growth and development, including the CAT-box for the regulation of the meristem, and the circadian, circadian discipline, and GCN4-motif for the regulation of the endosperm. Approximately 67.9% of the genes had at least one element involved in stress, including LTR for low temperature response, MBS for drought response, TC-rich repeats for stress response, and MBSI for flavonoid synthesis. Furthermore, 58.0% of the genes had W-box motifs, and these genes might interact with each other. These *cis*-acting elements assist or act on WRKY TFs and form WRKY-involved plant regulatory networks.

### 3.6. Gene Ontology of AsWRKY Proteins

The highly diverse protein sequences outside the conserved WRKY domain suggest that the AsWRKY gene family is involved in multiple regulatory processes [65]. Thus, we predicted the putative function of AsWRKY proteins by gene Ontology analysis. The result showed all GO term divided into three categories: “Molecular function”, “Cellular component”, and “Biological process” (Appendix A and Appendix A). In the molecular category, a total of 69 genes were significant and enriched the 12 subcategories; these proteins mainly attributed the transcription regulatory region nucleic acid binding (GO:0001067), transcription cis-regulatory region binding (GO:0000976), sequence-specific double-stranded DNA binding (GO:1990837), DNA-binding transcription factor activity (GO:0003700), and DNA binding (GO:0003677). In the cellular component category, 58 genes were enriched in nine subcategories, including chloroplast (GO:0009507), intracellular anatomical structure (GO:0005622), intracellular membrane-bounded organelle (GO:0043231), intracellular organelle (GO:0043229), membrane-bounded organelle (GO:0043227), nucleus (GO:0005634), obsolete intracellular part (GO:0044424), organelle (GO:0043226), and plastid (GO:0009536). Interestingly, the biological process category contained the greatest number of genes, with 83 genes enriched in 205 subcategories; these genes were mainly involved in hormone regulation, development, growth, and abiotic or biotic stress response.

### 3.7. miRNA Prediction of AsWRKY Interaction

miRNA can directly bind to TFs and regulate the expression of genes [66]. At the post-transcriptional level, miRNAs can regulate the expression of various WRKY TFs to regulate various processes [67]. Therefore, the interaction miRNA of AsWRKY genes were predicted by a strict parameter, a total of 29 AsWRKYs were targeted by 16 miRNAs (Figure 8 and Appendix A). Two members of the miR169 family target five genes (*AsWRKY116*, *AsWRKY117*, *AsWRKY142*, *AsWRKY159,* and *AsWRKY160*). Two members of the miR664 family target four genes (*AsWRKY132*, *AsWRKY10*, *AsWRKY38*, and *AsWRKY8*), and the miRN7 family also targets four genes (*AsWRKY116*, *AsWRKY128*, *AsWRKY155*, and *AsWRKY106*). Another known and novel miRNA targets one or multiple genes. The regulation function of the miRNAs in the WRKY family needs additional research to explain.

### 3.8. Expression Profiling of Oat WRKY Genes by RNA-Seq

Given the extensive regulatory role of the WRKY family in plants, we investigated the expression patterns of all 162 AsWRKY genes in public transcriptome data derived from various treatments such as lemma development, seed vigor, salt, drought with silicon, phosphorus deficit, and drought stress (Appendix A). It should be noted that not all AsWRKY genes respond positively to oat growth and development or to adversity. For example, the results showed that gene expression varied with treatment, the number of genes that were not expressed was 1 (*AsWRKY4*), 9 (*AsWRKY116/117/140/4/78/79/80/106/108*), 11 (*AsWRKY32/43/44/67/86/105/137/143/145/146/147*), 7 (*AsWRKY67/86/32/136/145/44/146*), 2 (*AsWRKY143* and *AsWRKY67*), and 1 (*AsWRKY95*) in lemma development, seed vigor, salt, drought with silicon, phosphorus deficit, and drought stress, respectively. Other WRKY genes may be upregulated or downregulated depending on the treatment, time, and developmental stage. For example, water-sensitive cultivars had more WRKY genes that exhibited upregulated expression under drought stress (Figure 9D), especially under 6 and 24 h of stress. For drought and silicon treatment, exogenous silicon induces some genes to exhibit more significant upregulated or downregulated expression under drought, and thus may be important genes involved in silicon-induced drought response (Figure 9E). Under salt stress, with increasing exposure to stress, more WRKY genes were upregulated after 12 h and 24 h, and only some genes were upregulated under short-term stress (Figure 9C). The expression of WRKY genes was more interesting under phosphorus deficiency; 138 genes changed from upregulated to downregulated, and 23 genes changed from downregulated to upregulated under phosphorus stress (Figure 9F). WRKY genes also showed differential expression patterns in regulating lemma development. Compared with BY685 cultivar, OA1613 cultivar showed more upregulation of AsWRKY genes in both early and late lemma development (Figure 9A). Furthermore, AsWRKY genes also actively regulated seed vigor, and only a few genes were upregulated after 16 days of seed aging, while half of the genes were significantly upregulated after 32 days.

### 3.9. Expression Pattern of AsWRKY Genes in Abiotic Stress

To explore the expression levels of AsWRKY genes under different abiotic stresses (drought and salt), qRT-PCR analysis was performed on 16 genes from AsWRKY members. Overall, some AsWRKY genes were significantly inhibited by PEG or salt treatment (Figure 10). For example, *AsWRKY83/161/156/53* responded to PEG and *AsWRKY161/156/36/53* to salt. In addition, when multiple AcWRKY genes were induced by the same treatment, their expression levels increased or decreased with the increase of time, which could be summarized as early and late expressed genes. For example, *AsWRKY10/65/72/84/5/76/22/94/152/119* responsed to drought stress, and *AsWRKY10/84/65/72/4/152 76/5/9/119/24/83* to salt stress.

## 4. Discussion

Based on its extensive regulatory role, an increasing number of WRKY family genes have been identified in plants in recent years. Given the versatility of the WRKY genes, comprehensive identification and functional validation of the oat WRKY family are expected to enrich databases in the field of oat development and stress resistance. However, to our knowledge, this is the first study that has identified the WRKY gene in the oat genome. In this study, we identified 162 AsWRKY genes and analyzed their basic structure and expression; these genes were named *AsWRKY1* to *AsWRKY162* (Appendix A).

The WRKY family predates plant differentiation such as a single copy of two WRKY domains was found in the lower species *G. lamblia*, *D. discoideum*, and *C. reinhardtii* [30]. Multiple sequence alignment of the conserved domains of oat and Arabidopsis revealed large variations involving the WRKYGQK motif in 22 AsWRKY proteins. It is unclear whether differences in these proteins affect the stability of their binding sites to the W box and the function of downstream target genes, but for Arabidopsis, mutation of the W box results in the loss of *AtWRKY33* binding ability [68]. In addition, the loss of WRKY or zinc finger motif was observed in oat, and a similar phenomenon occurred in wheat [37]. Phylogenetic analysis revealed that the protein sequences of these domain deletions in oats are better clustered into groups IIc, IIe, or III.

The loss of a domain does not always result in negative consequences. Although gene function is lost or changed temporarily, it may cause the expansion of gene families once again. Of course, this process can be extremely lengthy. For example, group II (except II c) and III may be the WRKY domain deletion of the N-terminal in I. Interestingly, we also have a similar event in the WRKY family of oat. IIa and IIb are divided into the same subgroup, and the same is true for IId and IIe. Both have a high similarity with the IC WRKY domain, but the former is closer to IC. Therefore, both clades may be derived from IC. Interestingly, phylogenetic analysis of WRKY sequences from Arabidopsis and rice merged the four clades of group II into two new sister groups IIa + b and IId + e; this similar model hints that they could also have an evolutionary relationship. Group III is considered to be the youngest branch in evolution, and its zinc finger structure (C_2_HC) is significantly different from group II. Surprisingly, *AsWRKY131* contains two common WRKY motifs, but the zinc finger structure (C_2_HC) has undergone a great change, which is different from IN and IC, but closer to group III. Brand et al. [69] conducted a study on the evolutionary conservatism of WRKY family and found that only grass subgroups had similar events, however there was insufficient phylogenetic evidence to support the independent classification of a new member of group I. We hypothesize that *AsWRKY131* may be a novel subgroup of Group I or that it is intending to evolve into Group III. There is evidence that both groups IIc and I evolved from a IIc-like ancestral WRKY, and the members of IIc in this study are more similar to group IC. Therefore, we could speculate that group IIc may have differentiated from Group I, or they may have come from the same ancestor.

The WRKY also has other potential conserved sequences that influence its functional diversity [26], and analyzing of motifs and intron-exon patterns of full-length protein sequences facilitates the identification of divergent domains and further corroborates the results of phylogenetic analysis [70,71]. The results showed that the WRKY subfamily of the same branch had similar structures, and almost all WRKY proteins contained motif 1 and motif 2, which further confirm that WRKY at the C-terminal of group I is the evolutionary ancestor of II and III. Meanwhile, *AsWRKY73* contained motif 3 and motif 4, which also confirmed our previous conjecture. Exon-intron structural diversity and cleavage patterns significantly affected gene family number and expansion [72]; the number of introns in the AsWRKY gene varies from 0 to 9 and more extensive variations are observed among clades. Our study found that the number of introns is not proportional to the length of the gene; *AsWRKY155* is the longest but only has three introns. According to studies in rice, after segmental duplication, introns are lost faster than introns are added [73]. In oat, groups IIa, IIb, IId, IIe, and III had significantly fewer introns, indicating that these WRKY clades evolved later than other groups. Alternative splicing is widespread in eukaryotic plants. A comparison of the number of transcripts of genes in different species (Arabidopsis, rice, maize, and wheat) found that wheat and oat have more transcripts, showing that the two have more abundant variable splicing, however the 0-intron phase is the main variable splicing in oat and 1-intron phase in wheat [37].

Most plants have undergone an ancient whole-genome duplication event or polyploidy, which is a large-scale chromosome doubling [74]. Every duplication or doubling of the genome will leave indelible traces (such as loss, transfer, and recombination) on the chromosomes. For example, the tetraploid ancestors of oat were hexaploidized, with large segments of chromosome 6D splitting into different chromosomes and only a portion of the hexaploid chromosome remaining [57]. In this study, the distribution of the AsWRKY gene on each chromosome was lowest in chromosome 6D (*AsWRKY146* and *AsWRKY147*). After analyzing gene duplication events, we found 111 duplication WRKY gene pairs, including 108 segmental duplications and three tandem duplications, indicating that segmental duplication is the main means of oat WRK family expansion. In addition, all duplicated gene pairs were subjected to strong purifying selection during evolution, indicating that the functional limitation of oat WRKY mainly comes from purifying selection. Hexaploid cultivated oats had completed the differentiation of each subgenome as early as 16 Mya, much earlier than the subgenome differentiation time of common wheat about 7 Mya. In this study, only five duplication gene pairs occurred before the doubling of the whole genome, and all of these occurred in the A and D subgenomes; the rest of the duplicate gene pairs occurred after subgenome differentiation, and the evolution time was between 0.18 and 15.3 Mya. Gene collinearity analysis among different species showed that oat and wheat had the highest conservation, followed by rice and maize, and the lowest was Arabidopsis. This difference may be related to the genomic characteristics and evolution of oat, which has many similarities in evolutionary origin with wheat because they both are allohexaploid. Meanwhile, Arabidopsis is dicotyledonous, suggesting that the WRKY gene pair for oat may have arisen after the differentiation of dicotyledonous and monocotyledonous plants.

TFs specifically bind to *cis*-acting elements to regulate the expression of related target genes [37,75], and a variety of *cis*-acting sites determine the diversity of regulatory functions of TFs [76]. In this study, AsWRKY genes were abundantly enriched in hormone-responsive and light-responsive modules such as AREB, CGTCA-motif, TGACG-motif, and G-box. In addition, most AsWRKYs contained the W-box core sequence TGAC/T, nearly half of the WRKY gene promoter regions contained the W-box in *Moringa oleifera* [77], and the DoWRKY target gene promoter region contained at least three W-box of *D. candidum* [78]. This indicates that there is cross-regulation between target genes and TFs, and their interaction or cooperation may also be the key to the extensive regulation of WRKY family. Gene annotations involving oat growth and development, hormone synthesis, and multiple biotic and abiotic stress response pathways further confirmed this idea.

miRNAs play an important role in repressing and cleaving genes at the post-transcriptional level. *Md-miR156ab* and *Md-miR395* could regulate *MdWRKYN1* and *MdWRKY26* in the response to fungal pathogens in *M. domestica* [79]; *tae-mir159a* was observed to target wheat WRKY gene under heat stress [80]; *Ha-miR396* targeted *WRKY6* to regulate the early responses to high temperature in *H. annuus* [66]; moreover, the main role of miR159 was the WRKY TFs [81]. In the current study, there was not a simple one-to-one relationship between miRNAs and AsWRKYs. One miRNA may target multiple genes, and the same WRKY gene may contain multiple miRNA targeting sites. Interestingly, some miRNA-targeted genes were specific, such as *miRNA169* and *miRN83*, which specifically targeted IIa and III. However, it remains to be considered whether a miRNA targeting genes in multiple WRKY subgroups, in addition to indicating that they share the same target, can indicate that they have a certain relationship in evolution.

TFs often regulate plant development and stress in the form of families, so that they can quickly respond to environmental changes [24,82]. Considerable evidence indicates that WRKY TFs respond to a variety of biological processes, such as *AtWRKY25/26/33/40/57/63* in Arabidopsis, *OsWRKY1/2/5/7/11/43/45* in rice, *GmWRKY13/21/54* in soybean, and *TaWRKY2/10/19* in wheat [21,83,84,85,86,87,88,89,90,91]. To better further validate the various regulatory responses that AsWRKY genes are involved in, we analyzed public transcriptomic data and qRT-PCR analysis on developmental or stress responses of different treatments. Almost all AsWRKY genes have participated in lemma development, drought stress, and phosphorus deficiency, but the expression patterns of these genes are not completely consistent in transcriptome data. Meanwhile, some genes showed different responses to drought or salt stress in qRT-PCR, and this difference may be related to external pressure selection during the evolution of the WRKY family.

## 5. Conclusions

In this study, we identified 162 AsWRKY genes from the whole oat genome. Phylogenetic relationships, gene structure, conserved motifs, chromosomal locations, promoter regions, and transcriptome expression were systematically analyzed. Amplification of the WRKY gene family in oat mainly occurred by segmental duplications, and most genes evolved after chromosome hexaploidy. The analysis of AsWRKYs function and expression pattern showed that most AsWRKY genes are actively involved in various processes of oat growth, development, and stress response, which provides extensive information for the continued study of AsWRKY genes.

## Figures and Tables

**Figure 1 genes-13-01918-f001:**
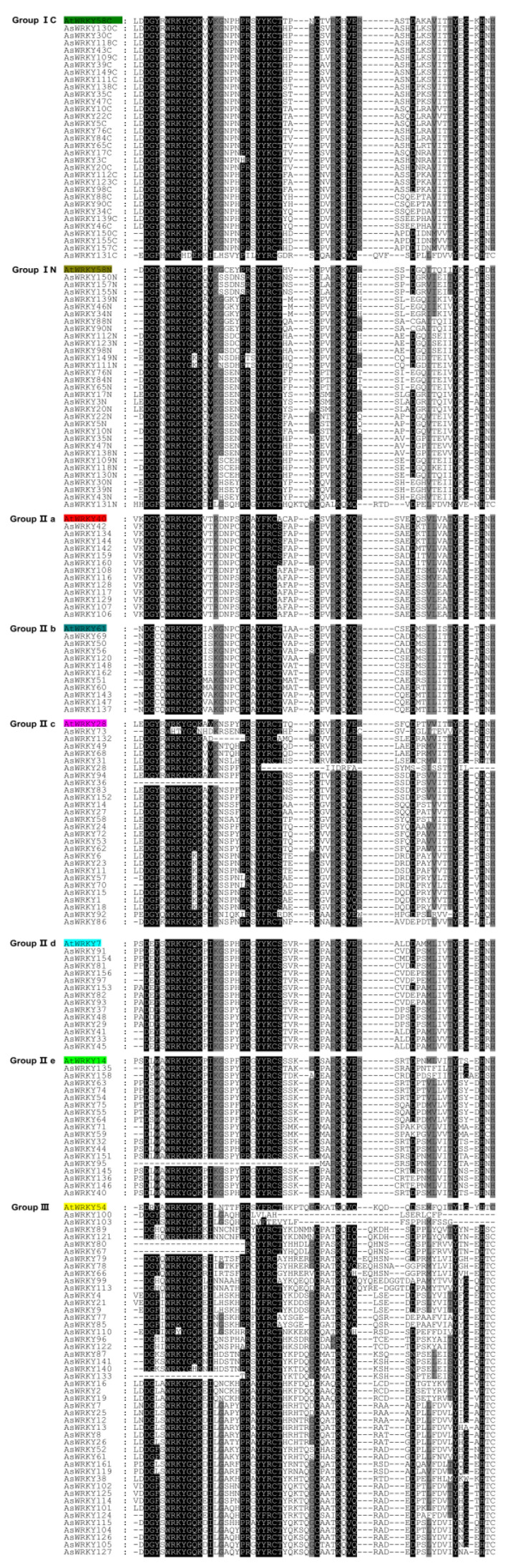
Multiple sequence alignments of AsWRKY and selected AtWRKY genes.

**Figure 2 genes-13-01918-f002:**
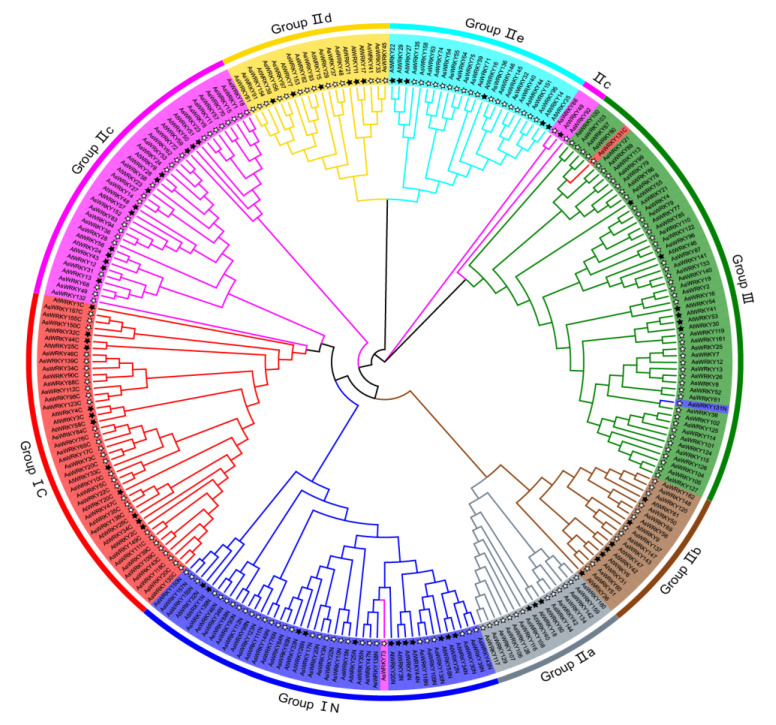
Phylogenetic relationship of AsWRKY domains among the Arabidopsis and oat. Different colored arcs are used to distinguish different groups or subgroups. The black solid star and hollow star represent Arabidopsis and oat, respectively.

**Figure 3 genes-13-01918-f003:**
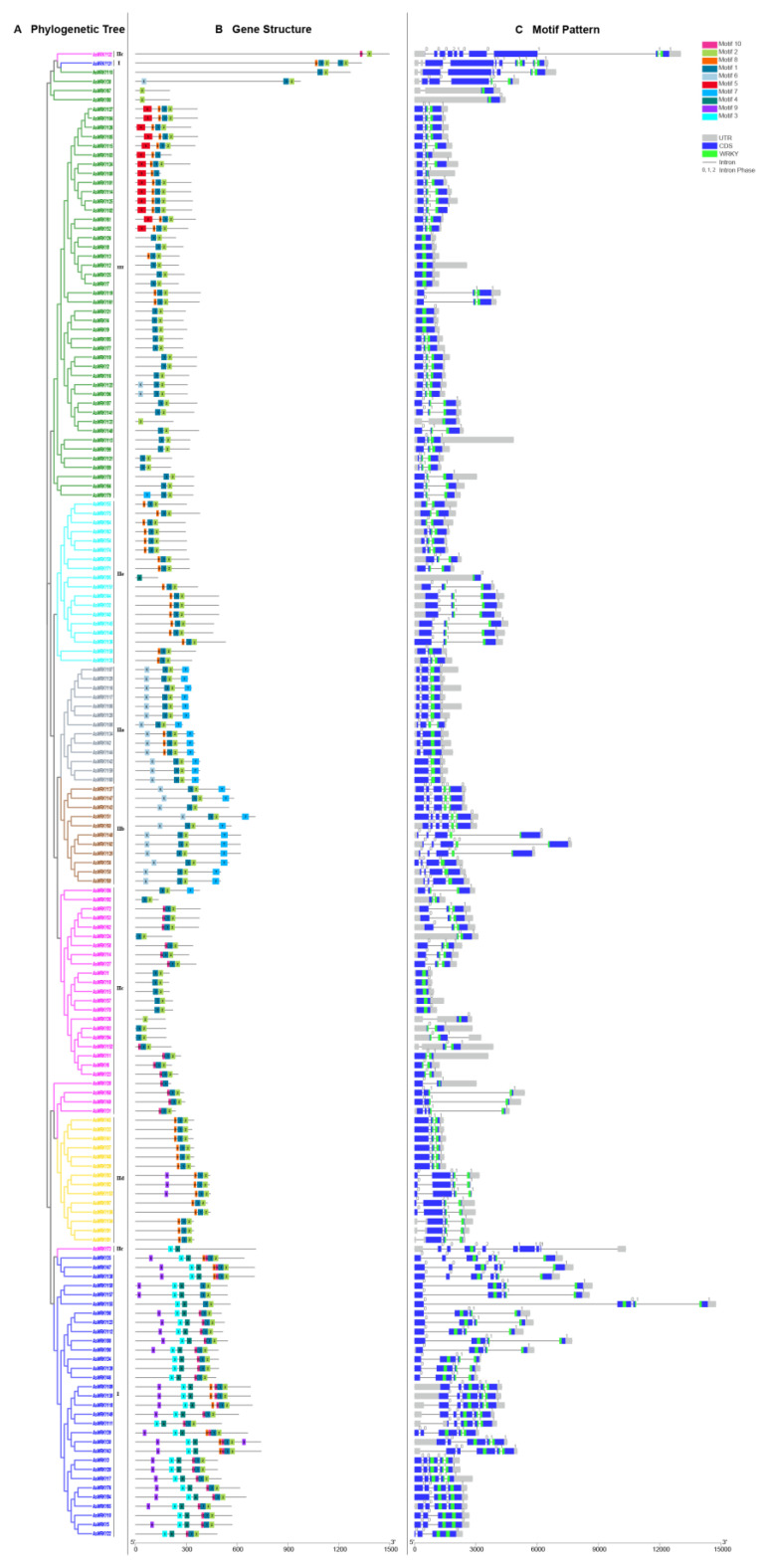
Phylogenetic relationship, conserved motifs and gene structure of AsWRKY proteins in oat. (**A**) The phylogenetic tree was constructed based on 162 AsWRKY full-length proteins using MEGA 7 software. (**B**) The conserved motifs, numbers 1–10, are shown in different colored boxes. The details of each motif provided by Appendix A. (**C**) The exon-intron structure of AsWRKY genes. Grey boxes represent 5′ UTR and 3′ UTR, blue boxes represent exons, greens boxes represent AsWRKY genes, and black lines present introns.

**Figure 4 genes-13-01918-f004:**
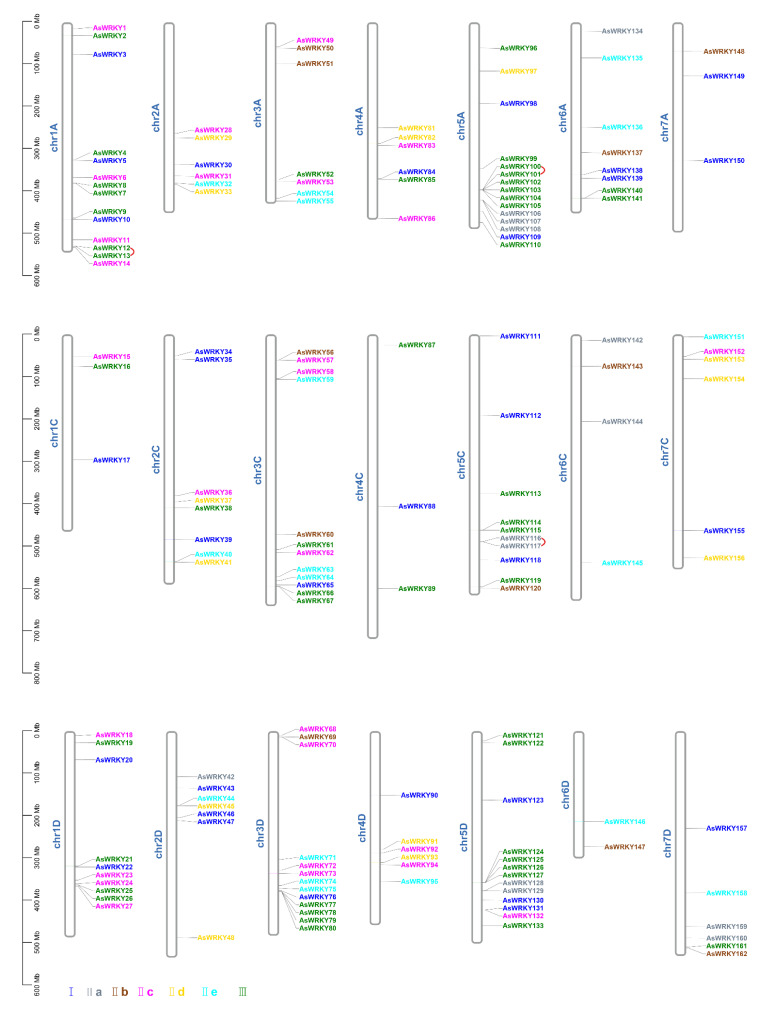
Chromosomal location of AsWRKY genes.

**Figure 5 genes-13-01918-f005:**
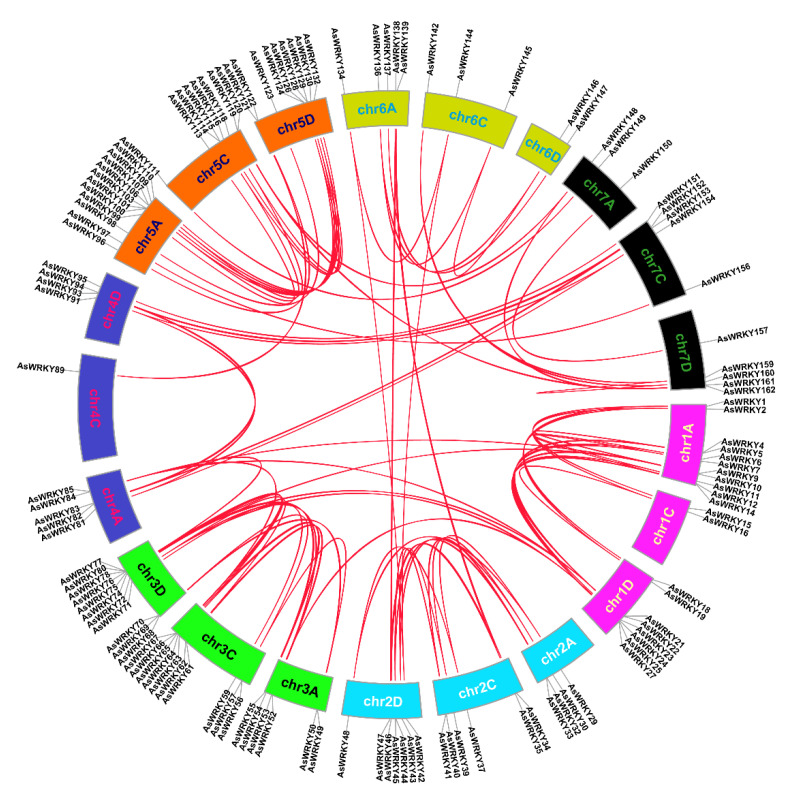
Synteny analysis of AsWRKY genes in oat. The red lines represent segmental duplication gene pairs. Chromosomes 1–7 are shown in different colors.

**Figure 6 genes-13-01918-f006:**
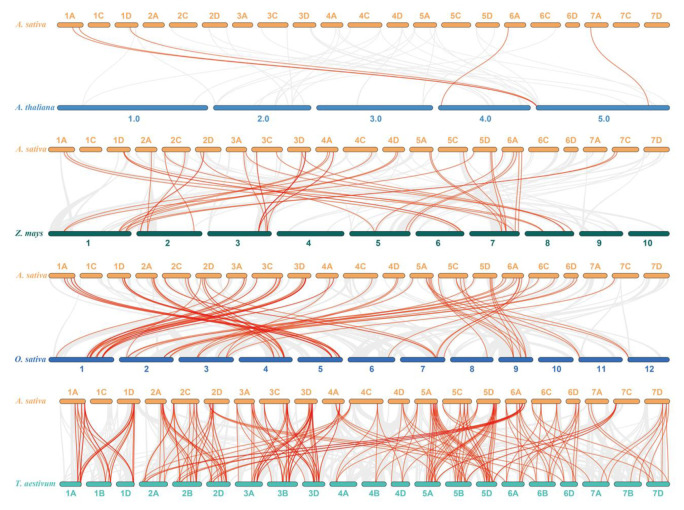
Synteny analysis of AsWRKY genes between oat and five representative plant species. The gray lines show colinear blocks in the genomes of oat with other plants, while the red line highlights the colinear WRKY pairs. The species name with the prefixes ‘*A. sativa*’, ‘*A. thaliana*’, ‘*Z. mays*’, ‘*O. sativa*’ and ‘*T. aestivum*’ indicate *Avena sativa*, *Arabidopsis thaliana*, *Zea mays*, *Oryza sativa*, and *Triticum aestivum*, respectively.

**Figure 7 genes-13-01918-f007:**
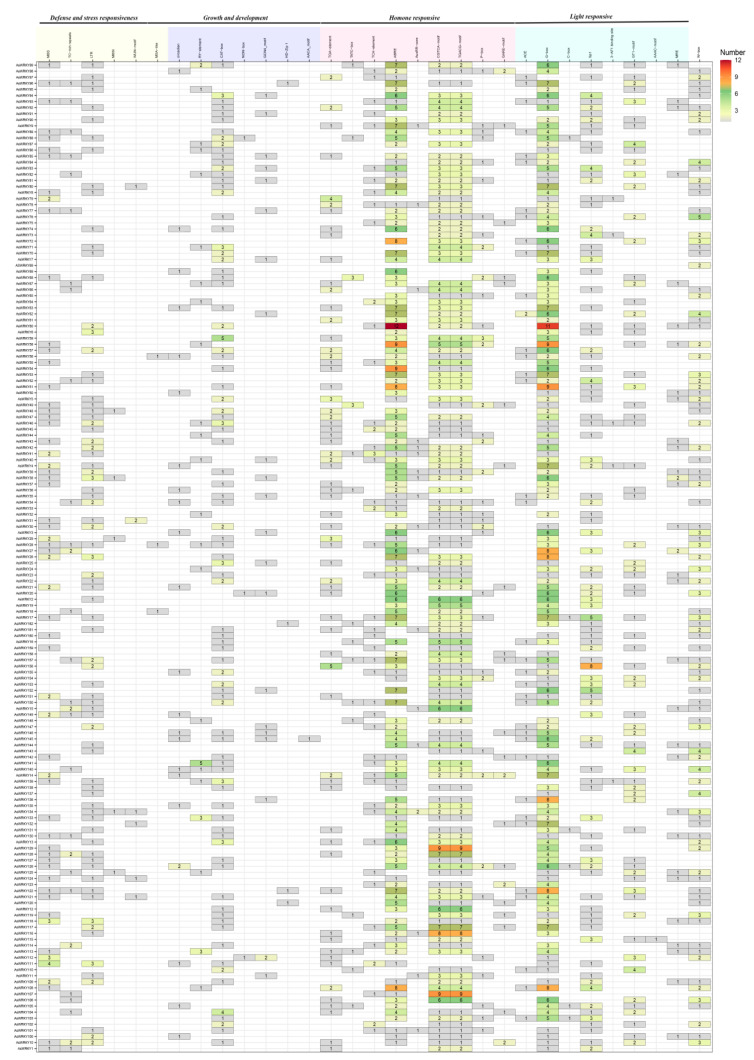
The *cis*-element in the promoter of AsWRKY genes. 3-AF1 binding site, light responsive element; AAAC-motif, light responsive element; AACA-motif, involved in endosperm-specific negative expression; ABRE, cis-acting element involved in the abscisic acid responsiveness; ACE, cis-acting element involved in light responsiveness; AuxRR-core, cis-acting regulatory element involved in auxin responsiveness; CAT-box, cis-acting regulatory element related to meristem expression; C-box, cis-acting regulatory element involved in light responsiveness; CGTCA-motif, cis-acting regulatory element involved in the MeJA-responsiveness; circadian, cis-acting regulatory element involved in circadian control; GARE-motif, gibberellin-responsive element; G-box, cis-acting regulatory element involved in light responsiveness; GCN4-motif, cis-regulatory element involved in endosperm expression; GT1-motif, light responsive element; HD-Zip 1, element involved in differentiation of the palisade mesophyll cells; LTR, cis-acting element involved in low-temperature responsiveness; MBS, MYB binding site involved in drought-inducibility; MBSI, MYB binding site involved in flavonoid biosynthetic genes regulation; MRE, MYB binding site involved in light responsiveness; MSA-like, cis-acting element involved in cell cycle regulation; NON-box, cis-acting regulatory element related to meristem specific activation; P-box, gibberellin-responsive element; RY-element, cis-acting regulatory element involved in seed-specific regulation; Sp1, light responsive element; TATC-box, *cis*-acting element involved in gibberellin-responsiveness; TCA-element, *cis*-acting element involved in salicylic acid responsiveness; TC-rich repeats, *cis*-acting element involved in defense and stress responsiveness; TGACG-motif, *cis*-acting regulatory element involved in the MeJA-responsiveness; TGA-element, auxin-responsive element; WUN-motif, wound-responsive element.

**Figure 8 genes-13-01918-f008:**
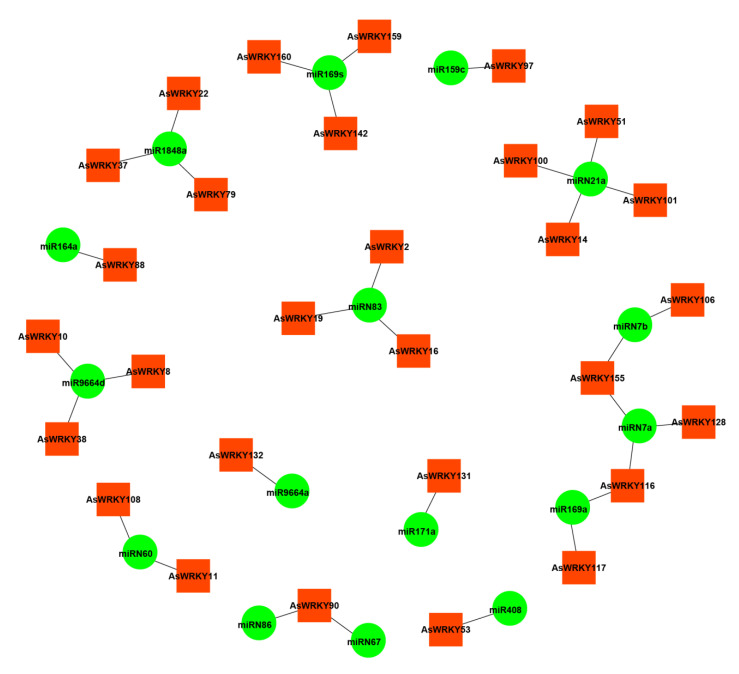
miRNA-AsWRKY prediction.

**Figure 9 genes-13-01918-f009:**
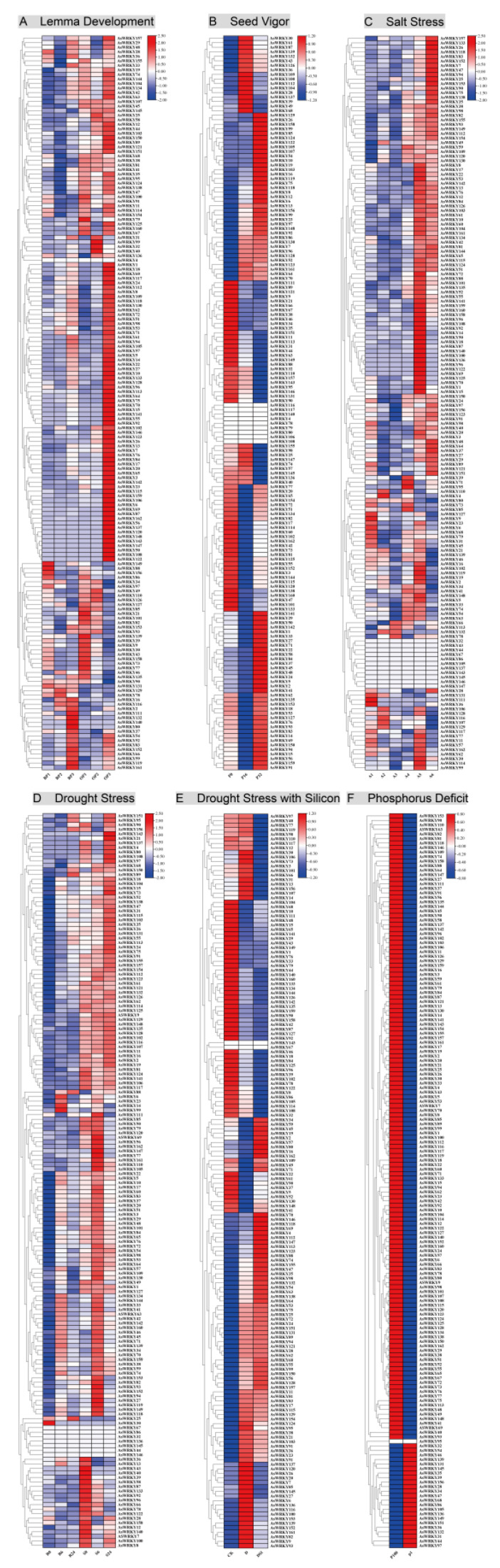
Expression pattern of 162 AsWRKY genes in different development stage or adversity based on RNA-seq data. (**A**) The comparison of lemma development between BY685 and OA1613 varieties at the P1, P2, and P3 stage. (**B**) The comparison of seed vigour after seeds aged 0 days (P0), 16 days (P16) and 32 days (P32). (**C**) Gene expression analysis of oat under salt stress at 0 h (A1), 2 h (A2), 4 h (A3), 8 h (A4), 12 h (A5), 24 h (A6). (**D**) The comparison of drought stress between DY2 and MW varieties at 0 h, 6 h, and 24 h. (**E**) WRKY gene response induced by exogenous silicon addition under drought stress. (**F**) The comparison genes expression for WRKY at 100 μM (P100) and 1 μM (P1) KH_2_PO_4_.

**Figure 10 genes-13-01918-f010:**
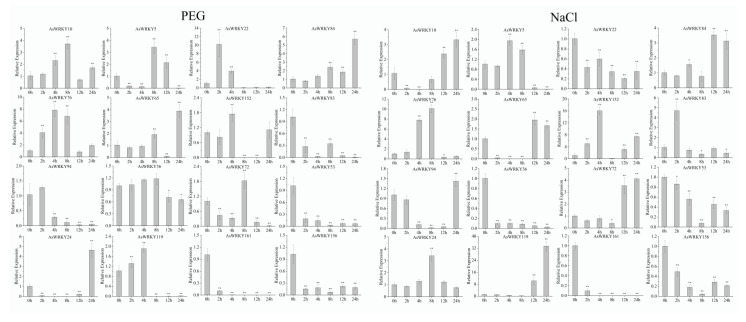
Expression pattern of 16 selected AsWRKY genes in response to drought and salt stress. Expression pattern of 16 selected AsWRKY genes in response to drought and salt stress. Significant differences were indicated as * (LSD test, *p* < 0.05) and ** (*p* < 0.01) between treatments and control.

## Data Availability

All data supporting the findings of this study are included in the article.

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
