# Peer review of "Genome-Wide Identification and Characterization of the Oat (*Avena sativa* L.) WRKY Transcription Factor Family"

_genes, 2022, doi:10.3390/genes13101918_

Round 1
Reviewer 1 Report
The paper provides a very comprehensive study of the WRKY TF family from the recently sequenced oat genome. The study is well conducted and may eventually become publishable in Genes after major revisions.
The most important point upon revision is the section “Data availability” which does not give any information yet. Here the authors should give links for the 162 AsWRKYs where all the information published in the submitted paper can be found. Such a link could also be a link to a genome database were the information can be extracted by submitting the assigned numbers of AsWRKY factors such as AsWRKY1 through AsWRKY162. This could be similar to the TAIR database for Arabidopsis thaliana. Alternatively, and strongly recommended, the authors could submit their data to the Plant Transcription Factor database at http://planttfdb.gao-lab.org. Currently no oat TFs have been annotated there.
Another very important point is the legibility of the text in the figures. The figures are almost illegible and need to be improved significantly. The supplementary figures are of much higher quality.
Here are some more points the authors should consider upon revising their paper.
L8/9 “The WRKY family is widely……, and is one…..”
L12 “…based on a genome-wide analysis in oat,…”
L13 “…a phylogenetic tree”
L14 “…gene duplication..”
L15 “111 gene pairs”
L17 ”functional analysis….that AsWRKYs are widely..”
L30 “They also act…”
L33-36 Citation is missing
L46 “C2HC”
L47 “, and were also”
L41-51 Rewrite this section because the way it is written it implicates that group IV was overlooked because of errors. However, in the cited publication the authors clearly state: “group IV genes might contain annotation errors caused by genomic sequencing or gene prediction software, or they might be pseudogenes with no biological functions”
Therefore, it is suggested to remove the statement that there is a group IV.
L53-55 Citation is missing
L53 “,TGAC is the core..” This statement should be modified since sequences containing the GACTTT core sequence of WT-boxes are also bound by WRKY factors. PubMed IDs 24104863, 28341984, 29076543, 33006643
L58 „almost all“
L61/62 This is the first time the involvement of WRKYs in plant defense responses is mentioned. However, this is a major function of WRKY TFs and should have been introduced much earlier in the introduction.
L60-65 This part needs to be rewritten. What are the “related genes” in different WRKY mutants? AtWRKY8 and AtWRKY28 are NOT in Botrytis cinerea and AtWRKY8 and AtWRKY48 are NOT in Pseudomonas syringae. It is suggested to write “AtWRKY8 and AtWRKY28 positively enhance resistance to Botrytis cinerea” and “AtWRKY8 and AtWRKY48 did not positively enhance resistance to Pseudomonas syringae”
L68 Citation is missing
L78 “The WRKY family already evolved during…”
L91 From the sentence it is not clear that the numbers in parenthesis are probably the number of WRKY family members in each species
L104 Genomes? “…from the sequences oat genome…”
L111 “…for the conserved…”
L112 “…these encode…”
L113 What does “gradation” mean?
L114 Could the authors also give the numbers of WRKY factors with one WRKY domain, not only with two.
L127 The “random selection of Arabidopsis WRKY factors” sounds awkward. Those should represent all known WRKY groups and probably do as shown in figure 1. Therfore, they are not randomly selected.
L130 “…that is the binding site..”
L130/131 “Various other WRKY domains were also….”
136 Insert space before “zinc”
L146 “…AsWRKY131 is highly homologous ….”
L148 „underwent“
L159 What are „conserved mods“?
L168 „..was higher than…“
L171 „..phases 1 and phase (?) in… “
L176 „The details of each motif..“
L192 The mapping data should be added in a supplementary data file with precise chromosomal location in relation to the oat genome sequence.
L196 “..greater numbers.”
L197 “..lowest number..”
L197/198 “..was also the shortest chromosome…”
L203 “Gene duplication….”
L242 “The species name…”
L246 The detection of putative cis-regulatory sequences is not a functional analysis. Rewrite the first sentence: “To identify potential cis-acting elements ….“.
L248 „PlantCARE“
L249 „…abiotic and biotic stress..“
L250-263 It is very surprising that the authors did not mention any WRKY binding sites in the promoters because also cis-sequences with the TGAC core were identified.
L288 “…that the AsWRKY gene family is….”
L292 “…of 69 genes were significant enriched in the 12 subcategories…”
L296“..enriched in 9…“
L301 What does „the abundance of biological process genes “ mean?
L302 „…enriched in..“
L305 miRNA regulation of gene expression by miRNA binding to promoters has, to my knowledge, not been shown in plants. The cited paper is a non-plant paper. Could the authors please cite a relevant paper on this from plants. Otherwise, this section is very speculative and should be revised by deleting miRNA/promoter events and by introducing the chapter in a more general way describing miRNA mediated degradation of transcripts and by concentrating on miRNA transcript interactions only in the chapter. It is very important to learn from this chapter which miRNAs bind in antisense orientation to the (spliced) WRKY transcripts and therefore have a potential function.
L331 Delete “materials”, treatment is enough
L342/343 Figure 9 only identifies AsWRKYs, the designation OA1613 and BY685 are not in the figure, although the figure text is almost illegible. Which AsWRKYs are OA1613 and BY685?
L348 BY685 and OA1613?
L351 What are DY2 and MW?
L356 How were those 16 genes selected?
L371 “..WRKY genes..”
L372/373 “….In this study, we identified 162…..and analyzed their basic structure and expression. These genes were….“
L382 See comment regarding group IV above
L395 “…also have an…”
L396 “…branch in evolution…”
L399 What do you mean by “conduced on ”?
L410 „WRKY“
L419/420 Citation is missing for rice example
L427 „carriers of gene replication“? A very uncommon expression
L447 “…are allohexaploid“
L455 Give the scientific name of Drumstick
L483/484 What does “were special response” means?
L500 “…were subjected to…”
L525 “….protein sequences were…”
L530 “AsWRKY genes”
L583 Provide a data availability statement. See comment above
L591 References. Check a references carefully according to journal style. A few examples that need to be corrected or edited follow.
L598 Journal name is missing
L600 Journal name is incomplete
L621 Journal name is incomplete
L630 Journal name is missing
L634 Journal name needs to be edited
L676 Journal name is incomplete
L698 Journal name is missing
L751 Journal name needs to be edited
Figure S1 Intron
Reviewer 2 Report
In this manuscript authors have identified the 162 WRKY transcription factors in oat (Avena sativa) using genome-wide approach and reported the unevenly distribution across 21 chromosomes. Authors have annotated the identified members into three groups namely, I, II, and III based on the phylogeny analysis. Overall the manuscript is interesting and well written, however, I have following comments for authors to improve the quality of MS.
1. In title and MS, authors are advise to use the full scientific name of oat as Avena sativa L.
2. In Abstract, define the how many genes you have selected for the qRT-PCR and provide the name of genes showing significant expression profiling.
3. Provide the flow chart of methodology for better understanding of concept.
4. In methods, cite the all tools and web-servers in significant manners, only URLs are not sufficient.
5. All genes name should be as per nomenclature and must be in italic.
6. In introduction, provides the more significant background of WRKYs, authors my utilize following article https://link.springer.com/article/10.1007/s11103-018-0761-6, and https://link.springer.com/article/10.1007/s00122-016-2794-z
7. Predict the gene specific SSRs in genomic sequences of identified genes.
8. Provide the physichemical properties of identified WRKYs, authors may utilize ggplot2 for plot visualization.
9. Predict the 3D structure model of few WRKYs.
10. What was the basis to select the genes for qRT-PCR?
11. Calculate the phylogeny tree with WRKYs from more plant species
Round 2
Reviewer 2 Report
Authors have significantly addressed my all comments, I don't have more comments. Therefore, I would recommend this MS for publication.